# Influence of *Acacia Mearnsii* Fodder on Rumen Digestion and Mitigation of Greenhouse Gas Production

**DOI:** 10.3390/ani12172250

**Published:** 2022-08-31

**Authors:** Luis Vargas-Ortiz, Veronica Andrade-Yucailla, Marcos Barros-Rodríguez, Raciel Lima-Orozco, Edis Macías-Rodríguez, Katherine Contreras-Barros, Carlos Guishca-Cunuhay

**Affiliations:** 1Departamento de Producción Animal, Instituto Superior Tecnológico Benjamín Araujo, Patate 180403, Ecuador; 2Centro de Investigaciones Agropecuarias, Universidad Central “Marta Abreu” de Las Villas, Santa Clara 50100, Cuba; 3Centro de Investigaciones Agropecuarias, Facultad de Ciencias Agrarias, Universidad Estatal Península de Santa Elena, La Libertad 240204, Ecuador; 4Facultad de Ciencias Agropecuarias, Universidad Técnica de Ambato, Sector el Tambo-La Universidad, vía a Quero, Cevallos 1801334, Ecuador; 5Facultad de Ciencias Veterinarias, Universidad Técnica de Manabí, Portoviejo 130701, Ecuador; 6Facultad de Ciencias Biológicas, Universidad Central del Ecuador, Campus El Dorado-Itchimbía, Quito 170403, Ecuador

**Keywords:** *acacia mearnsii*, rumen degradation, tannin, methane

## Abstract

**Simple Summary:**

The anthropogenic generation of greenhouse gases (GHG) from the production of ruminants has contributed to environmental deterioration throughout the world; therefore, reducing their production becomes one of the main objectives today. The manipulation of the ruminant diet with forage sources rich in bioactive compounds (tannins) is considered an alternative to mitigate the production of CH_4_ and improve the productive performance of the animals. Based on this, this research aimed to evaluate the effect of the incorporation of different amounts of *Acacia mearnssi*, rich in tannins, on the parameters of ruminal degradation, digestibility of DM and OM, and the generation of gas, CH_4_, and CO_2_. The incorporation of *A. mearnssi* forage in the ration had a reducing effect on GHG production and possibly improved utilization of dietary protein in response to the presence of tannins. However, even with the lowest level of acacia in the diet, the effective digestion of DM and OM was affected. Under the conditions of this study, it was concluded that it is possible to replace traditional forages with up to 20% of *A. mearnsii*, without observing changes in the production of greenhouse gases with respect to the control treatment (0% of *A. mearnsii*); however, *A. mearnsii* is not usable because it significantly decreases rumen degradability of DM and OM, which would considerably affect the production in animals.

**Abstract:**

In recent years, the worrying generation of GHG from ruminant production has generated widespread interest in exploring nutritional strategies focused on reducing these gases, presenting the use of bioactive compounds (tannins) as an alternative in the diet. The aim of this research was to determine the effect of the addition of different levels of *Acacia mearnsii* on ruminal degradation, nutrient digestibility, and mitigation of greenhouse gas production. A completely randomized design with four treatments and six repetitions was used. The treatments were: T1, T2, T3, and T4 diets with, respectively, 0%, 20%, 40%, and 60% *A. mearnsii*. The rumen degradation kinetic and in vitro digestibility, and the production of gas, CH_4_, and CO_2_ were evaluated. In situ rumen degradation and in vitro digestibility of DM and OM showed differences between treatments, with T1 being higher (*p* < 0.05) in the degradation of the soluble fraction (A), potential degradation (A + B), and effective degradation for the different passage rates in percent hour (0.02, 0.05, and 0.08), compared to the other treatments. Rumen pH did not show differences (*p* > 0.05) between treatments. The lowest (*p* < 0.05) gas, CH_4_, and CO_2_ production was observed in treatments T1 and T2 with an approximate mean of 354.5 mL gas/0.500 g fermented DM, 36.5 mL CH_4_/0.500 g fermented DM, and 151.5 mL CO_2_/0.500 g fermented DM, respectively, compared to treatments T3 and T4. Under the conditions of this study, it was concluded that it is possible to replace traditional forages with up to 20% of *A. mearnsii*, without observing changes in the production of greenhouse gases with respect to the control treatment (0% of *A. mearnsii*); however, *A. mearnsii* is not usable because it significantly decreases rumen degradability of DM and OM, which would considerably affect the production in animals.

## 1. Introduction

Ruminant production systems in tropical and subtropical regions are generally based on grass monocultures. The nutritional value of these grasslands ranges between 7 and 10% protein and its content of structural carbohydrates exceeds 70% [1,2]. These factors affect feed digestibility probably due to the imbalance generated between the fermentable organic matter in the rumen and the availability of nitrogen (N) that hinders the growth and activity of ruminal microorganisms [2,3]; on the other hand, it promotes greenhouse gas (GHG) in response to the decrease in the microbial protein synthesis capacity and the high fiber concentration of the forage [4]. Methane (CH_4_), carbon dioxide (CO_2_), and nitrous oxide (N_2_O) are the main GHGs generated by anaerobic fermentation in the rumen [5], causing considerable energy losses in the animal, ranging from 2–12% of the consumed energy [6,7], which causes environmental deterioration and decreased animal production [8,9]. It is estimated that livestock production generates about 14.5% of GHGs worldwide [10]. In this context, the FAO [11] calculates that ruminants under grazing conditions generate approximately 47% of CH_4_ with respect to the total GHGs from livestock production.

The most viable alternative to counteract these drawbacks has been the manipulation of rumen fermentation through the diet, using forages, agro-industrial by-products or seeds that, due to their high nutritional value and contribution of bioactive compounds (tannins, saponins, essential oils) can favorably modulate the ruminal environment and thereby reduce GHGs of enteric origin [12,13,14]. Under these perspectives, it has been shown that the use of nonconventional forage sources rich in bioactive compounds, especially tannins, can improve rumen fermentation and with it, digestion. These effects allow to reduce the production of gases generated enterically, and, consequently, increase in the productive performance of the animals [15]. These benefits can be attributed to: (i) the ability of tannins to form tannin–protein complexes and maximize nitrogen utilization in the lower parts of the gastrointestinal tract [16], (ii) decreased CH_4_ generation due to the suppression of methanogenic archaea and protozoa [17,18], and (iii) anthelmintic control of gastrointestinal parasites [19,20]. As a consequence of the above, greater milk production and daily weight gain (10–21% and 8–38%, respectively) have been observed when feeding sheep and goats with forages rich in tannins, probably due to the greater flow of metabolizable protein towards the parts from the gastrointestinal tract in response to decreased protein degradation in the rumen [21]. This has encouraged exploring the usefulness of tannins, focused on reducing GHGs generated from ruminant production and improving productivity [22]. However, the secondary compounds could have unwanted effects on the digestibility of the feed, probably due to the formation of complexes with carbohydrates in the diet [23].

The effects of tannins can vary depending on the dose, source, type, molecular weight (MW) [24], and adaptability of ruminants to their consumption [25]. Saminathan et al. [26] studied the effect of the MW of the CT of *Leucaena leucocephala* hybrida and showed a lower production of total gas, CH_4_, and CO_2_ and better utilization of N without negative effects on the digestibility of the DM with the highest MW. However, negative effects of tannins on consumption, digestion, productive performance, and health have been noted, probably attributed to: (a) lower feed palatability due to the binding of tannin to salivary glycoproteins [27], (b) decreased nutrient digestibility and consequent reduction in feed transit in the rumen [21], (c) toxicity with high levels of tannins (>55 g CT/kg DM) [28] with the consequent deterioration of the intestinal mucosa, liver, and kidney [29], and (d) decreased intestinal activity of pancreatic enzymes (trypsin, amylase) and absorption of amino acids [30]. Robins and Broker [31] evidenced a reduction in feed intake and body weight in sheep fed with CT of *Acacia aneura* (7.5 g/100 g DM) due to the formation of ulcers in the abomasum. Henke et al. [32] observed a decrease in the nutritional components of milk (protein and fat) as a result of the high consumption of CT provided by the quebracho extract (3.0 g/100 g DM vs. 1.5 g/100 g DM) given by the decrease in protein digestibility in the rumen. Garg et al. [33] reported toxic effects on the renal and hepatic system due to the consumption of *Quercus incana* tannins in cattle without prior adaptation and subsequent manifestation with hematochezia and edema in the ventral region of the thorax with a mortality margin of 70%. Effects were mainly associated with the inadaptability of ruminants to the consumption of forage sources rich in tannins in their ration [34].

Forages rich in tannins have been studied previously and the lack of consistency in the results on GHG production [35,36,37] requires more research to be carried out in order to find the balance point for their use in ruminant feed without promoting harmful effects on their health [38]. *A. mearnsii* is a plant distributed throughout the world [39]. The genus Acacia is widely distributed in tropical and warm temperate regions, it is made up of approximately 1350 species with diverse uses, previously reviewed by Correia et al. [40]. *A. mearnssi* adapts to temperatures that fluctuate between 14.7–27.8 °C, characterized by the presence of CT (35–45%) [41]. In recent years, *A. mearnssi* has been promoted as a source of tannin extracts to be used in ruminant feed for methane reduction [42]. That there is only limited information about *A. mearnssi* as a forage source which could be a limitation for its use in ruminant feeding [39], but the species belongs to the same genus as *Acacia cyanophylla* which has been shown to be useful as a forage source in small ruminants. This is probably in response to the nutritional content [protein (12.2% DM), NDF (42.1% DM), ADF (36.1% DM), and TC (31.5 g/kg DM)]. Addition of 100 g/day of acacia, corresponding to 2.88 g CT in dairy sheep feed improved the digestibility of CP, NDF, OM and decreased the excretion of N in the urine [43]. Based on this background, the objective of this research was to determine the effect of the addition of different levels of *A. mearnsii* on ruminal degradation, nutrient digestibility, and mitigation of gas, CH_4_, and CO_2_ production.

## 2. Materials and Methods

### 2.1. Study Location

The present investigation was carried out at “Querochaca” Experimental Farm and Rumenology Laboratory of the Universidad Técnica de Ambato, Facultad de Ciencias Agropecuarias, Tungurahua, Ecuador, at an altitude of 2890 m above sea level. In the sector, there are maximum temperatures of 20 °C and minimum of 7 °C and an average ambient temperature of 15 °C.

### 2.2. Animals

Six three-year-old Holstein bulls with an average live weight of 450 ± 21.2 kg, provided with a fistula with a cannula in the rumen (Bar Diamond, Parma, ID, USA) were used. The animals were housed in individual pens with a zinc roof and cement floor and access to a diet based on 50% *Medicago sativa* + 50% *Lolium perenne* and ad libitum water.

### 2.3. Forage Samples and Treatments

The *A. mearnsii* forage was collected from a two-year-old plantation located at the Faculty of Agricultural Sciences—UTA (abbreviation in Spanish), subjected to a cutting frequency of 90 d. Subsequently, the forage (leaves and young stems: 50 kg) was dehydrated under cover in a greenhouse. The dehydrated forage was ground in a hammer mill to a particle size of 2 mm and proceeded to be incorporated in the following treatments (Table 1). Six repetitions were performed for each treatment (*n* = 6). Prior to mixing the treatments, the forages were separately passed through a 1 mm sieve to homogenize the particle size.

### 2.4. Ruminal Degradation Kinetic

In situ ruminal degradation of nutrients was estimated following the nylon bag methodology (0.42 µ) in the rumen described by Ørskov et al. [44]. In each bull (*n* = 6), a bag with 5 g of each diet was incubated at the following times (hours): 3, 6, 12, 24, 48, 72, and 96 h. At the end of 96 h, the bags were removed, washed with running water and dried at 60 °C. The residues were stored in polyethylene bags at −4 °C until their subsequent analysis in the laboratory. Nutrient disappearance was calculated as a ratio of incubated and residual material. The data was fitted to the equation: Y = a + b (1 − e^−ct^), and the effective degradation was fitted using the equation DE = a + [(b * c)/(c + k)] considering passage rates (*k*) of 0.02, 0.05, and 0.08% [45], (Prisma 4, GraphPad Software, Inc. San Diego, CA, USA).

### 2.5. Gas, CH_4_, CO_2_ Production and In Vitro Digestibility

Rumen content (liquid and solid fraction) was obtained separately from each bull (*n* = 6). The ruminal content was collected before feeding in the morning and stored in plastic containers, and then transported to the laboratory to be processed within the first hour of collection. The preparation of nitrogen-rich media (artificial saliva) was performed as described by Menke and Steingass [46]. Gas, CH_4_, and CO_2_ production was established using the methodology described by Theodorou et al. [47] which consists of placing 0.5 g samples of each of the treatments T1, T2, T3, and T4 in amber glass bottles with a capacity of 100 mL. Sixty mL of the inoculum (70:30 medium; artificial saliva/inoculum; ruminal content) were incubated in the bottles under a constant flow of CO_2_. The bottles were incubated between 39–40 °C. The measurement of gas pressure and volume was taken manually at the following times: 3, 6, 9, 12, 18, 24, 36, 48, 60, 72, and 96 h after incubation with a pressure transducer (DO 9704, Delta OHM, Casella, Italy) and plastic syringes. The CH_4_ and CO_2_ production was quantified with a GAS Detection analyzer, model GX-6000, UK following the methodology described by Elghandour et al. [48]. For each treatment, six bottles were used, and three additional bottles were used as blanks. At the end of 96 h, the data were fitted to the monobasic equation mLgas = GV (1 + (B/t) ^C^)^−1^ described by Groot et al. [49]. Additionally, six more flasks for each treatment were incubated for up to 48 h to estimate the in vitro digestibility of DM and OM [47]. Gas, CH_4_, and CO_2_ data was reported in mL/0.500 g fermented DM.

### 2.6. Rumen pH

Under the same procedure mentioned above for gas production and digestibility, 6 amber glass flasks were prepared for each treatment and at each time (6, 12, and 24 h postincubation) ruminal pH was measured with the help of a pH meter (BANTE-221 Portable pH/ORP Meter, London, UK).

### 2.7. Chemical Analysis

The dry matter (DM) (# 7007) and ash (# 7009) were determined according to the AOAC [50]. Neutral detergent fiber (NDF) and acid detergent fiber (ADF) were determined using method 12 and 13, respectively, ANKOM2000 fiber analyzer (ANKOM Technology, Macedon, NY, USA). CP was determined by elemental analysis (N) using a LECO CHN 628 (LECO Corporation, Michigan, USA). Condensed tannins were determined by vanillin assay (catechin equivalent, Price et al. [51]).

### 2.8. Experimental Design and Statistical Analysis

A completely randomized design was used, with four treatments and six repetitions. All variables were analyzed according to the design used by means of a simple classification ANOVA [52]. Means were compared using Tukey’s test. Additionally, surface response analysis was carried out to assess the linear, quadratic, or cubic effects [52] of the response to treatments. All variables were analyzed using the SAS (version 9.2, SAS Institute, Cary, NC, USA).

## 3. Results

### 3.1. Rumen Degradation and Digestibility of DM and OM

In situ ruminal degradation (Table 2) and in vitro digestibility (Table 3) of DM and OM showed a descending linear effect (*p* = 0.0001) as the inclusion level of *A. mearnsii* in the diet increased, with degradation being the higher (*p* = 0.0001) of the soluble fraction (A), potential degradation (A + B) and effective degradation for the different passage rates in percent hour (0.02, 0.05 and 0.08) in T1, with respect to the other treatments. Ruminal pH did not show differences between treatments in any of the evaluated hours (6 h, 12 h, and 24 h) (*p* = 0.7078, 0.3319 and 0.8729, respectively) (Table 3). 

### 3.2. Gas, CH_4_, and CO_2_ Production

Gas production (*p* = 0.0001), CH_4_ (*p* = 0.0004), and CO_2_ (*p* = 0.0063) showed differences between treatments. T1 and T2 showed the lowest values with an approximate mean of 354.5 mL gas/0.500 g fermented DM, 36.5 mL CH_4_/0.500 g fermented DM, and 151.5 mL CO_2_/0.500 g fermented DM, respectively, compared to treatments T3 and T4. Showing a linear effect (*p* < 0.05) to the response of the treatments (Table 4). Gas and CO_2_ production kinetics (Figure 1A,C, respectively) are observed to start at 3 h with a notable increase until hour 96. However, the production of CH_4_ in T3 and T4 started at 12 h (Figure 1 B) and in T1 and T2 from approximately 20 h, reaching stabilization in all treatments at 72 h (Figure 1B).

## 4. Discussion

### 4.1. Rumen Degradation and Digestibility of DM and OM

The effect of tannins on degradation, digestibility, ruminal fermentation, and CH_4_ production are closely related to dose, type, source, and molecular weight [24]. In this context, the linear decrease (Table 2) in the potential degradation (A + B), effective degradation, and digestibility (Table 3) of DM and OM observed in T2, T3, and T4 is probably due to the increase of tannin in the diets (Table 1) as a consequence of the incorporation of *A. mearnsii* and its effect on fiber degradation as a possible response to the formation of tannin–cellulose complexes, reduction of cellulitic microorganisms, and inhibition of the binding capacity of fibrolytic microorganisms on the substrate to be degraded [18]. These are mechanisms that can affect feed intake and productive performance of animals, attributed to the enzymatic inactivity responsible for fiber degradation and subsequent decrease in the rate of feed passage in the rumen [21,53]. Similar results were reported by Kozloski et al. [54], who found lower nutrient digestibility when supplying 20, 40, and 60 g/kg of *A. mearnsii* tannin in sheep feed. Contrary to this, Avila et al. [42] found no negative effects of the inclusion of tannin from *A. mearnsii* on the digestibility of DM, OM, and NDF when using doses lower than 20 g/kg DM in the feeding of steers. Something observed in this research, the evident relationship between the content of tannin in the diet and its effect on digestion (Table 2). The ruminal pH reported in this study (Table 3) are consistent with those reported by de Oliveira et al. [55] and Hariadi and Santoso [56], who did not show pH changes in the ruminal fluid due to the addition of tannins and indicated that they are in the optimal range to maintain a balanced cellulolytic activity (pH: 6.7 ± 0.5).

### 4.2. Gas, CH_4_, and CO_2_ Production

The parameters of gas, CH_4_, and CO_2_ production observed in T1 and T2 (Table 4 and Figure 1) are probably due to the higher digestion obtained (Table 3) and, on the other hand, to the lower contribution of dihydrogen (H_2_) used as a substrate for methanogenesis and released in the process of formation of acetic acid from pyruvate [57]. This highlights the direct correlation between the chemical composition of the feed, the digestibility of DM and OM, and accumulation of ruminal H_2_ and volatile fatty acids (VFA) [58]. These results are consistent with those reported by Makkar [53] and Aragadvay-Yungán et al. [15]. However, the results obtained in treatments T3 and T4 of this study were possibly due to the lower degradation and digestibility of nutrients (Table 2) as an effect of the higher proportion of acacia in the diets. The larger proportion of *A. mearnsii* resulted in the greater content of tannins (Table 1), and these results are consistent with those reported by Kelln et al. [59]. In addition, the higher production of gas, CH_4_ and CO_2_ evidenced in T3 and T4 (Table 2) could be linked to the low digestibility of the fiber as a direct effect of tannin [18], which probably interrupted the binding capacity of the microorganism to the cell wall of the plant and, consequently, inhibited the action of microbial enzymes useful for the degradation of the fibrous component of the substrate [22,60]. This probably decreases the availability of protein located in the cell wall of the forage (fibrolized protein) and, consequently, reduces the ability to synthesize microbial protein [58]. In this context, Blümel et al. [58] showed an inversely proportional relationship between the synthesis of microbial biomass and the volume of gas produced. Evidenced in the present study with a higher production of gas, CH_4_, and CO_2_ as the digestibility of the diet decreased (Table 4). These results are consistent with those reported by Henke et al. [32] and Barros-Rodriguez et al. [61].

## 5. Conclusions

Under the conditions of this study, it was concluded that it is possible to replace traditional forages with up to 20% of *A. mearnsii*, without observing changes in the production of greenhouse gases with respect to the control treatment (0% of *A. mearnsii*); however, *A. mearnsii* is not usable because it significantly decreases rumen degradability of DM and OM, which would considerably affect the production in animals.

## Figures and Tables

**Figure 1 animals-12-02250-f001:**
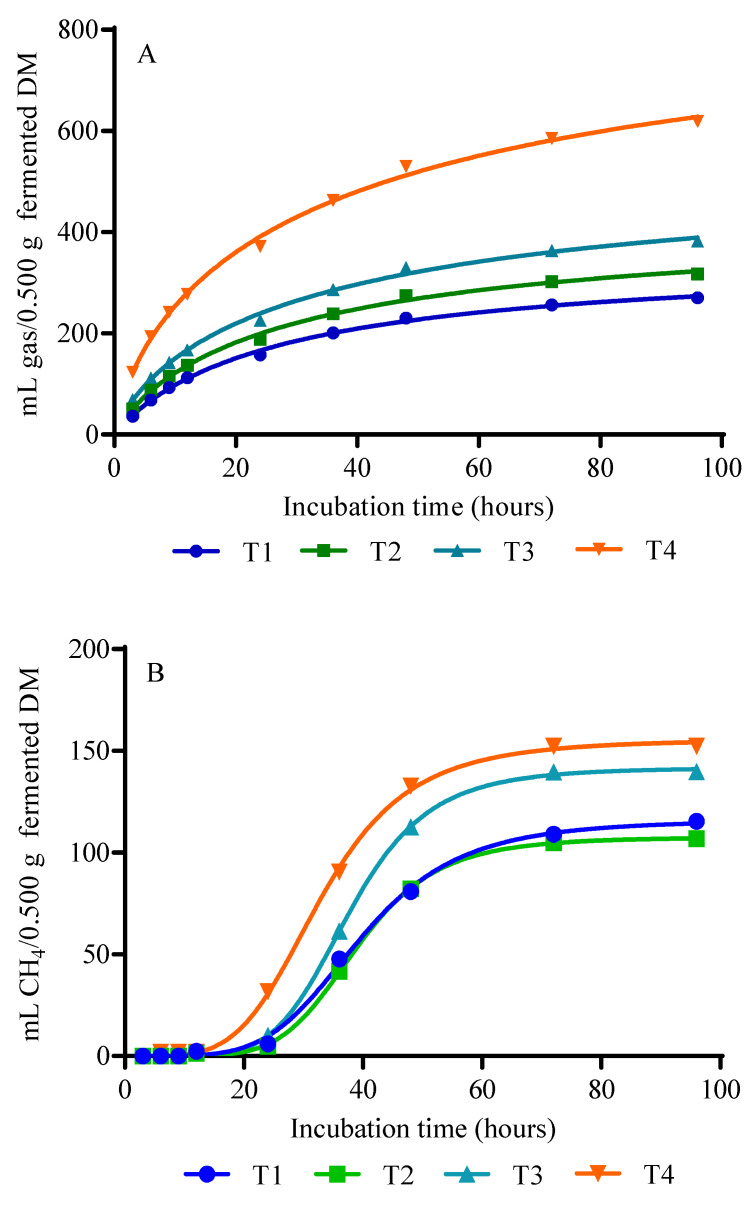
Gas (**A**), CH_4_ (**B**), and CO_2_ (**C**) production kinetics of diets with increasing levels of *A mearnsii*. T1: diet 0% inclusion of *A. mearnsii*, T2: diet 20% inclusion of *A. mearnsii*, T3: diet 40% inclusion of *A. mearnsii,* T4: diet 60% inclusion of *A. mearnsii.*

**Table 1 animals-12-02250-t001:** Experimental treatments and chemical composition (% except where otherwise noted) of diets with increasing levels of *Acacia mearnsii.*

Items	Treatments			
	T1	T2	T3	T4
Forage meal (*A. mearnsii*)	0.00	20.00	40.00	60.00
Forage meal (*L. perenne*)	66.45	54.28	42.10	29.93
Forage meal (*M. sativa*)	33.55	25.72	17.90	10.07
Chemical composition				
Dry matter	88.67	89.37	90.08	90.80
Organic matter	90.39	91.21	92.04	92.86
Crude protein	19.00	20.10	20.01	21.21
Ether extract	3.51	3.35	3.19	3.03
Neutral detergent fiber	43.77	41.27	40.77	39.28
Acid detergent fiber	23.54	23.94	24.34	24.74
Metabolizable energy (MJ/kg DM)	9.21	9.20	9.20	9.19
Condensed tannins	0	3.56	6.03	7.97

T1: diet 0% inclusion of *A. mearnsii*, T2: diet 20% inclusion of *A. mearnsii*, T3: diet 40% inclusion of *A. mearnsii*, T4: diet 60% inclusion of *A. mearnsii.*

**Table 2 animals-12-02250-t002:** In situ rumen degradation kinetics and in vitro digestibility (DM and OM) of diets with increasing levels of *A.*
*mearnsii* (except where otherwise noted).

	Treatment				SE	*p*-Value	Contrasts		
	T1	T2	T3	T4			L	Q	C
Degradation DM									
A	45.6 ^a^	40.3 ^b^	34.7 ^c^	29.6 ^d^	1.16	0.0001	0.0001	0.9033	0.9142
B	44.1 ^a^	40.0 ^a^	37.5 ^a^	37.4 ^a^	1.85	0.0641	0.0144	0.2652	0.8983
*c*	0.05 ^a^	0.05 ^a^	0.04 ^a^	0.04 ^a^	0.007	0.3210	0.0777	0.7802	0.6712
A + B	89.8 ^a^	80.2 ^b^	72.1 ^c^	67.1 ^c^	2.01	0.0001	0.0001	0.2718	0.8570
Effective Degradation *									
0.02	77.0 ^a^	68.1 ^b^	58.7 ^c^	51.0 ^d^	0.94	0.0001	0.0001	0.5360	0.5790
0.05	67.8 ^a^	59.6 ^b^	50.6 ^c^	43.6 ^d^	1.06	0.0001	0.0001	0.5583	0.5504
0.08	62.9 ^a^	55.1 ^b^	46.6 ^c^	40.1 ^d^	0.96	0.0001	0.0001	0.5039	0.5245
Degradation OM									
A	43.2 ^a^	38.3 ^b^	32.6 ^c^	29.0 ^c^	1.19	0.0001	0.0001	0.5896	0.5824
B	46.5 ^a^	41.5 ^a^	38.7 ^a^	38.9 ^a^	2.27	0.0810	0.0195	0.2631	0.9205
*c*	0.06 ^a^	0.05 ^a^	0.04 ^a^	0.04 ^a^	0.01	0.2513	0.0517	0.8550	0.7429
A + B	89.7 ^a^	79.8 ^b^	71.2 ^b. c^	67.9 ^c^	2.45	0.0001	0.0001	0.1977	0.7189
Effective Degradation *									
0.02	76.7 ^a^	67.6 ^b^	57.8 ^c^	50.5 ^d^	0.95	0.0001	0.0001	0.3247	0.4577
0.05	67.1 ^a^	58.7 ^b^	49.3 ^c^	43.0 ^d^	1.08	0.0001	0.0001	0.3634	0.4193
0.08	61.8 ^a^	54.0 ^b^	45.2 ^c^	39.5 ^d^	0.98	0.0001	0.0001	0.2926	0.3563

^a–d^ Means with different letter between rows differ significantly (*p* < 0.05). T1: diet 0% inclusion of *A. mearnsii*, T2: diet 20% inclusion of *A. mearnsii*, T3: diet 40% inclusion of *A. mearnsii*, T4: diet 60% inclusion of *A. mearnsii*. A: degradation of the soluble fraction, B: degradation of the insoluble but potentially degradable fraction, *c*: degradation rate in % per hour, A + B: degradation potential. *: effective degradation at ruminal passage rates of 2, 5, and 8% for hour. DM: dry matter, OM: organic matter, L: linear contrast, Q: quadratic contrast, C: cubic contrast, SE: standard error.

**Table 3 animals-12-02250-t003:** In vitro digestibility (DM and OM) of diets with increasing levels of *A.*
*mearnsii* (%, except where otherwise noted) and pH.

	Treatment				SE	*p*-Value	Contrasts		
	T1	T2	T3	T4			L	Q	C
Digestibility									
DM	74.8 ^a^	58.1 ^a b^	41.5 ^b^	20.7 ^c^	4.57	0.0001	0.0001	0.9115	0.7292
OM	77.1 ^a^	59.5 ^a b^	41.5 ^b^	20.6 ^c^	4.93	0.0001	0.0001	0.6918	0.9360
pH									
6 h	6.98 ^a^	7.00 ^a^	6.98 ^a^	6.97 ^a^	0.02	0.7078	0.5613	0.3746	0.6364
12 h	6.95 ^a^	7.00 ^a^	7.00 ^a^	7.02 ^a^	0.03	0.3319	0.1009	0.5161	0.6338
24 h	7.26 ^a^	7.23 ^a^	7.28 ^a^	7.29 ^a^	0.06	0.8729	0.5607	0.7940	0.6048

^a–c^ Means with different letter between rows differ significantly (*p* < 0.05). T1: diet 0% inclusion of *A. mearnsii*, T2: diet 20% inclusion of *A. mearnsii*, T3: diet 40% inclusion of *A. mearnsii*, T4: diet 60% inclusion of *A. mearnsii*. DM: dry matter, OM: organic matter, L: linear contrast, Q: quadratic contrast, C: cubic contrast, SE: standard error.

**Table 4 animals-12-02250-t004:** Gas, CH_4_, and CO_2_ production parameters (mL/0.500 g fermented DM) of diets with increasing levels of *A mearnsii.*

	Treatment				SE	*p*-Value	Contrasts		
	T1	T2	T3	T4			L	Q	C
Gas production									
GV (mL)	360.9 ^c^	434.7 ^c^	561.3 ^b^	944.1 ^a^	20.80	0.0001	0.0001	0.0001	0.0404
B	28.9 ^a^	29.4 ^a^	35.8 ^a^	38.6 ^a^	3.29	0.1340	0.0268	0.7162	0.5274
*c*	1.0 ^a^	0.9 ^a b^	0.8 ^b c^	0.8 ^c^	0.02	0.0001	0.0001	0.8372	0.7898
CH_4_ production									
GV (mL)	116.7 ^b c^	107.7 ^c^	141.8 ^a b^	156.1 ^a^	7.26	0.0004	0.0001	0.1249	0.0671
B	39.9 ^a^	39.1 ^a^	37.8 ^a^	33.2 ^b^	1.07	0.0013	0.0003	0.0975	0.5768
*c*	4.8 ^a^	6.0 ^a^	5.7 ^a^	5.1 ^a^	0.41	0.1938	0.7180	0.0412	0.6001
CO_2_ production									
GV (mL)	170.9 ^b^	189.7 ^b^	229.7 ^a b^	425.4 ^a^	49.75	0.0063	0.0017	0.0905	0.5528
B	108.8 ^a^	105.9 ^a^	106.5 ^a^	132.5 ^a^	33.50	0.9315	0.6377	0.6709	0.8856
*c*	1.1 ^a^	1.0 ^a^	1.0 ^a^	0.9 ^a^	0.05	0.2739	0.0674	0.6085	0.6846

^a–c^ Means with different letter between rows differ significantly (*p* < 0.05). T1: diet 0% inclusion of *A. mearnsii*, T2: diet 20% inclusion of *A. mearnsii*, T3: diet 40% inclusion of *A. mearnsii*, T4: diet 60% inclusion of *A. mearnsii*. GV, B, and *c*: the parameters of the mL gas equation CH_4_ or CO_2_ = GV (1 + (B/t)^C^)^−1^ (Groot et al., 1996) (see text). L, linear contrast; Q, quadratic contrast; C, cubic contrast; SE, standard error.

## Data Availability

The data presented in this paper are available on request from the corresponding author.

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
