# Peer review of "Influence of Acacia Mearnsii Fodder on Rumen Digestion and Mitigation of Greenhouse Gas Production"

_animals, 2022, doi:10.3390/ani12172250_

Round 1

Reviewer 1 Report

Influence of Acacia mearnsii Fodder on Rumen Digestion and

Mitigation of Greenhouse Gas Production

After reading the manuscript “Influence of Acacia mearnsii Fodder on Rumen Digestion and Mitigation of Greenhouse Gas Production” (reference number 1733755), I recommend publication in the journal after minor revision.

Simple summary

I suggest following the authors’ guidelines and rewriting this section. A simple summary must show the following: “the aims and objectives, pertinent results, conclusions from the study and how they will be valuable to society”. Therefore, stick to the guidelines and add some relevant results from your study.

Abstract

L26 Include a background sentence before the objective highlighting the purpose of the experiment.  

L28 Include the animal characteristics used in the study, i.e., sex, weight, breed, and age.

Keywords

There are some repeated keywords in the title. Then, I suggest changing them to different keywords.

Introduction

L45-47 Specify the zone or climatic conditions for those grass characteristics. Also, I suggest changing the reference by a review because that paper was based on leguminous species.

L47 Which factors? I agreed that high structural carbohydrates. However, I suggest expanding the explanation about low CP content increasing the GHG.

L52-53 Include an actual estimation of CH4 on the GHG, that reference described the same value (40%) from Gerber et al., 2013.

L54-55 Include also some disadvantages of manipulating diets with plants rich in secondary compounds, i.e., the reduction in feed digestibility.

L70-71 Include a sentence explaining why you focused on tannins.

L77-78 Change the reference to others using ruminants; that paper did not correctly explain the phenomenon. Several papers with sheep, goats, deer, etc., show the relationship between saliva proteins and secondary compounds.

L82-90, I suggest finishing the sentence with a link idea mention that those adverse effects also depend on the adaptability of the ruminants to secondary compounds, as you mentioned in L72.

L83 For the first time, Include the complete scientific name of Acacia aneura.

L91-93 Add which kind of results? According to the review, it was on the methane mitigation strategies.

L93-96 These three lines are not enough to justify the use of Acacia mearnsii as a model for the study. Expand the justification of using A. mearnsii as a candidate for your experiment. In addition, change “excellent nutritional characteristics” by results from papers on similar climatic conditions compared with your study and include CP, EM, fibres, digestibility, and CT contents to let the readers judge if the plant has an optimal nutritional value analysis.

Materials and methods

I did not find how long the study lasted.

L102-104 Include the climatic characteristics of the study area.

L109 Explain the proportions of each feed on the basal diet.

L111-118 How the other forage meals were prepared?

L146-147 Include the reference for the in vitro DM and OM digestibility.

L162-166 Include the normality and homogeneity of variances tests of the dataset. Also, add the software and the procedures used to analyse the dataset.

Results

L171 Include the exact P-value according to the test. The author’s guidelines stated, “significance tests are required to report exact p values and effect sizes for all inferential analyses.”

L179-181 Some T1 and T2 results shared literals with T3 (Ch4 production and Co2 production). Therefore, change the writing to “T1 and T2 showed some of the lowest values….”

How did you calculate those means?

Discussion

L189-190 In the results, it was estimated the CT was catechins. Any idea about other secondary compounds that could affect the mitigation of CH4, for example, saponins from Acacia mearnsii.

How can you explain the absence of statistical difference between T1 and T2 for DM and OM digestibility (Table 3)?

Conclusions

Rewrite the conclusion based on the results of the study. There was no real reduction in GHG because T1 as a control showed a similar parameter as T2. On the other hand, increasing the A. mearnsii inclusion showed an opposite effect in the study.

Tables

Tables 2,3, and 4 Change between rows by within rows.

Table 3 changes a,b,c, and d by a-c superscripts. There is no d in the table

Change the literals format to superscript format

Minor issues

L182 Delete the space before the < symbol

Table 4 title Change 05 g by 0.5g

L183 Change gas by Gas

L215 Describe the full name for the first time in the text.

L223 delete the. before to

Author Response

ALL SUGGESTIONS HAVE BEEN REALIZED

Reviewer 2 Report

The experiment is methodologically correct. However, the  results and conclusions are not true. The authors concluded that “ the incorporation of 20% of A. mearnsii in the ruminant diet decreases CH4 production without negative effects on nutrient digestibility”. However, according to the showed Tables, the inclusion of A. mearnsii did not decrease CH4 production or CO2, compared to T1 (Control). Even more, in situ degradation of DM and OM was lower when A. mearnsii was included at any level. The results show that the foliage of A. mearnsii did not reduce CH4 and CO2 production, and that dietary inclusion of A. mearnsii higher than 20% can affect digestibility of dry matter and organic matter.

The authors require to adequate the results and conclusions in consequence.

Author Response

ALL SUGGESTIONS HAVE BEEN REALIZED 

Reviewer 3 Report

Please, see the attached file

Author Response

ALL SUGGESTIONS HAVE BEEN REALIZED 

Round 2

Reviewer 2 Report

The conclusion still remains incorrect. It must say that Acacia mearnsii is not an option to mitigate greenhouse gases coming from rumen fermentation.

Author Response

The conclusion still remains incorrect. It must say that Acacia mearnsii is not an option to mitigate greenhouse gases coming from rumen fermentation,

the suggestion has been considered. suggested changes were made